# Epidemiological Evaluation of Events Allegedly Attributable to COVID-19 Vaccination: A Cross-Sectional Study in the Brazilian Amazon

**DOI:** 10.3390/ph17030304

**Published:** 2024-02-27

**Authors:** Matheus Sallys Oliveira Silva, Giovanni Moura Sotelo, Franciane de Paula Fernandes, Livia de Aguiar Valentim, Marcelo Silva de Paula, Tatiane Costa Quaresma, Márcia Jeane do Rego Dias, Géssica Aleane Moraes Esquerdo, Waldiney Pires Moraes, Sheyla Mara Silva de Oliveira

**Affiliations:** 1Health Department, University of the State of Pará, Av. Plácido de Castro, Santarém 68040-090, PA, Brazil; matheus.ssilva@aluno.uepa.br (M.S.O.S.); giovanni.sotelo@aluno.uepa.br (G.M.S.); franciane.fernandes@uepa.br (F.d.P.F.); marcellodipaula86@gmail.com (M.S.d.P.); tatiane.quaresma@uepa.br (T.C.Q.); jeanedias.10@hotmail.com (M.J.d.R.D.); sheyla.oliveira@uepa.br (S.M.S.d.O.); 2Institute of Public Health—(ISCO), Federal University of Western Pará (UFOPA), Santarém 68040-255, PA, Brazil; gmoraesesquerdo@gmail.com (G.A.M.E.); waldiney.moraes@ufopa.edu.br (W.P.M.)

**Keywords:** coronavirus infections, vaccines, adverse effects, epidemiological surveillance

## Abstract

The emergence of SARS-CoV-2, leading to the widespread outbreak of COVID-19, has unveiled a spectrum of symptoms and severe health complications, challenging healthcare systems and impacting millions of lives worldwide. To analyze events allegedly attributable to vaccination or immunization (ESAVI) against SARS-CoV-2 (COVID-19) in the municipality of Santarém, in the interior of the state of Pará, an epidemiological, descriptive study was conducted using data from e-SUS Notifica in Santarém/PA from January 2021 to January 2022. The analyzed data for ESAVI cases included the administered immunobiologicals (Coronavac, Covishield, and Comirnaty), the type of event, case progression, time in days between immunobiological administration, and symptom onset, causality, and classification of ESAVI according to the vaccine package inserts. The incidence rate of ESAVI due to the COVID-19 vaccine was 17 per 100,000 doses administered in the municipality. According to the ESAVI classification, 14.0% were classified as Serious ESAVI (ESAVIG) (IR: 8.12 per 100,000 doses administered), with 100% of these events resulting in full recovery without sequelae, and 82.4% of reported cases were classified as Non-Serious ESAVI (ESAVING) (IR: 47.78), of which 3.60% were immunization errors (IR: 2.08 IE per 100,000 doses). This study fosters discussion on the importance of accurate recording of ESAVI related to COVID-19 vaccines, demonstrating their safety for the population.

## 1. Introduction

The new coronavirus, designated as Severe Acute Respiratory Syndrome Coronavirus-2 (SARS-CoV-2), and known as Coronavirus Disease-19 (COVID-19) for the disease it causes, presents a range of symptoms that vary from individual to individual, ranging from a simple flu-like illness to severe pneumonia [1].

Since December 2019, the global community has been battling COVID-19, triggered by a new coronavirus. Originating in Wuhan, China, the virus swiftly crossed international borders, prompting outbreaks in several Asian countries, including Thailand, Japan, South Korea, and Singapore, before affecting Europe and other regions. This rapid spread compelled the World Health Organization (WHO) to announce a Public Health Emergency of International Concern by 30 January 2020, escalating to a pandemic declaration on 11 March 2020 [2,3].

Although the fatality rate of the disease caused by SARS-CoV-2 is lower than that of other coronaviruses, its high transmissibility has resulted in a higher number of deaths than the combined epidemics of SARS-CoV-2 and Middle East Respiratory Syndrome (MERS-CoV). The virus primarily spreads through contaminated droplets from an infected individual’s oropharyngeal secretions to an uninfected individual, as well as through contact with contaminated objects and surfaces, where it can remain active for up to 72 h [4].

The health crisis triggered by COVID-19, caused by the SARS-CoV-2 virus, has established itself as an unparalleled event in the history of public health, touching the lives of millions of people around the globe since its emergence at the end of 2019. The reach of its impact goes far beyond the cases directly related to the infection, affecting a wide variety of health conditions and challenging health systems worldwide. Current research highlights the complex relationship between COVID-19 and other pathologies, showing how the virus can exacerbate existing diseases, modify the incidence of certain conditions, and further pressure medical care systems that are in a vulnerable state. A clear example of this is the study that points to the effects of the pandemic on the mental and physical well-being of children, notably with an increase in the rates of childhood obesity, a result of changes in lifestyle and a reduction in physical activity [5].

Furthermore, COVID-19 has been shown to have significant implications for cardiovascular diseases, as evidenced in studies [6], which discuss the impact of COVID-19 on hypertension and cardiovascular complications. The research underscores the importance of a holistic approach to treating patients with COVID-19, considering coexisting conditions. Other studies [7,8] provide additional insights into the consequences of the pandemic in various areas, including its effect on the prevalence of infectious and non-infectious diseases, and the pressure on public health systems. These studies collectively underline the urgent need for integrated public health strategies to mitigate the impact of COVID-19 and protect vulnerable populations.

Given the global state of emergency, the World Health Organization (WHO), in collaboration with governments, scientists, pharmaceutical industries, and non-governmental organizations, has developed strategies for the development and production of a vaccine that can be offered as a global common good. Scientific and technological advancements, along with substantial knowledge gained about the immune response to the disease, have led to the creation of several vaccines that are now available for COVID-19 immunization campaigns [9,10].

The development of vaccines against SARS-CoV-2 has become synonymous with hope but the rapid manner in which they were produced, bypassing some steps of the process, has raised questions among the population regarding the risks and benefits in light of adverse reactions following vaccination. It is expected that any pharmaceutical product may produce some adverse effects after administration. What needs to be analyzed is the severity of these events [2,11,12].

ANVISA, the National Health Surveillance Agency in Brazil, is responsible for establishing, coordinating, and monitoring toxicological and pharmacological surveillance systems. It also regulates, controls, and oversees all vaccines administered to the population and monitors events allegedly attributable to vaccination or immunization (ESAVI) through notifications made in the e-SUS Notifica system. Pharmacovigilance aims to monitor, detect, evaluate, and prevent adverse events or any problems related to immunobiologicals, which can be reported by healthcare professionals as well as the general population [13].

While vaccination aims to contain a pandemic scenario, the population still experiences moments of insecurity regarding the safety of anti-COVID-19 vaccines. There is still much uncertainty among the population about the risks associated with vaccination and/or immunization. However, one way to address these doubts is to monitor adverse effects following immunobiological administration. Therefore, pharmacovigilance studies are of great importance in better understanding the situation. Thus, the objective of this study was to analyze ESAVI against SARS-CoV-2 (COVID-19) in the city of Santarém, Pará, Brazil.

## 2. Results

In Table 1, we present data on ESAVI notifications and their correlation with gender and doses administered in Santarém, Pará, from January 2021 to January 2022. Among females, the Coronavac vaccine from Sinovac/Butantan resulted in 13 notifications, with 34,053 doses administered and an incidence rate of 38.17 per 100,000 doses. The Covishield vaccine from AstraZeneca/Fiocruz had 27 notifications, 91,706 doses administered, and an incidence rate of 29.44. For the Comirnaty vaccine from Pfizer/Wyeth, there were eight notifications and 102,955 doses administered, yielding an incidence rate of 7.77. The total number of ESAVI notifications for females was 48, with an incidence rate of 20.99 per 100,000 doses.

Among males, the Coronavac vaccine had nine notifications, 29,660 doses administered, and an incidence rate of 30.34. Covishield had 13 notifications, 84,960 doses administered, and an incidence rate of 15.30. The Comirnaty vaccine had four notifications, 87,789 doses administered, and an incidence rate of 4.55. The total number of ESAVI notifications for males was 26, with an incidence rate of 12.85 per 100,000 doses. In summary, there were 74 ESAVI notifications, involving a total of 431,123 doses administered and resulting in a general incidence rate of 17.16 per 100,000 doses.

In Table 2, we provide key information regarding the classification of ESAVI notifications for different vaccines. Specifically, for the Coronavac vaccine from Sinovac/Butantan, 63,713 doses were administered, with no notifications classified as Severe ESAVI and 63 notifications classified as Non-Severe ESAVI. This translates to an incidence rate of 0.00 for Severe ESAVI and 98.88 for Non-Severe ESAVI. Similarly, for the Covishield vaccine from AstraZeneca/Fiocruz, there were 176,666 doses administered, resulting in 32 notifications classified as Severe ESAVI and 101 notifications classified as Non-Severe ESAVI. This led to an incidence rate of 18.11 for Severe ESAVI and 57.17 for Non-Severe ESAVI. Concerning the Comirnaty vaccine from Pfizer/Wyeth, 190,744 doses were administered, with three notifications classified as Severe ESAVI and 42 notifications classified as Non-Severe ESAVI, resulting in an incidence rate of 1.57 for Severe ESAVI and 22.02 for Non-Severe ESAVI. In total, considering all vaccines and a cumulative administration of 431,123 doses, there were 35 notifications classified as Severe ESAVI and 206 notifications classified as Non-Severe ESAVI, yielding an overall incidence rate of 8.12 for Severe ESAVI and 47.78 for Non-Severe ESAVI. Within the cases of EI, 22.22% were also linked to adverse events potentially attributed to vaccination or immunization, with an incidence rate of 0.46 per 100,000 doses administered. Out of a total of nine notifications classified as immunization errors, two cases (22.2%) were associated with ESAVI, while seven cases (77.7%) were not.

Out of the total notifications, 37.84% of affected individuals were between the ages of 18 and 35. Regarding the timing of administration and symptom onset, the majority of the population studied experienced symptoms on the same day as the vaccination (81.1%). Among the age groups, individuals aged 18 to 35 accounted for 28 notifications (37.8%), followed by those aged 36 to 49 with 27 notifications (36.5%), 8 notifications (10.8%) in the 50 to 64 age group, and 11 notifications (14.9%) in individuals over 64 years old.

Regarding the vaccination and symptom onset timeframe, 60 cases (81.1%) reported symptoms on the same day as the vaccination, while 5 cases (6.8%) experienced symptoms in the days following vaccination.

It is worth noting that the most commonly reported adverse effects across all vaccines were myalgia, fever, and pain at the injection site. Furthermore, the majority of ESAVI cases were associated with the first dose of the vaccines.

Among the most recurrent symptoms reported in the events, fever had an incidence rate (IR) of 27.13 per 100,000 doses administered for the first dose and 2.46 per 100,000 doses for the second dose. Headache was only reported in the first dose with an incidence rate of 27.13 per 100,000 doses. Myalgia had an incidence rate of 18.09 per 100,000 doses for the first dose and 1.23 per 100,000 doses for the second dose. Diarrhea had an incidence rate of 3.99 per 100,000 doses for the first dose and 2.46 per 100,000 doses for the second dose. Nausea and pain at the injection site were only reported in events related to the first dose of the vaccine, with incidence rates of 7.91 and 5.65 per 100,000 doses administered, respectively. No ESAVI notifications were related to booster doses of Covishield.

The most frequently reported events were a headache with an incidence rate (Ti) of 29.36 per 100,000 doses administered for the first dose, 3.38 per 100,000 doses for the second dose, and no reports for booster doses. This was followed by myalgia with an incidence rate of 5.87 per 100,000 doses for the first dose, 6.76 per 100,000 doses for the second dose, and 9090.9 per 100,000 doses for booster doses. Diarrhea had an incidence rate of 11.74 per 100,000 doses administered and 3636.4 per 100,000 doses for booster doses. Vomiting had an incidence rate of 14.68 per 100,000 doses for the first dose and 1818.2 per 100,000 doses for booster doses. Fever had an incidence rate of 5.87 per 100,000 doses for the first dose and 7272.7 per 100,000 doses for booster doses. Pain at the injection site had an incidence rate of 2.94 per 100,000 doses administered and 5454.5 per 100,000 doses for booster doses. None of these symptoms were reported during the second dose.

Among the vaccines administered in Santarém, Comirnaty, the vaccine produced by Pfizer/Wyeth had the lowest percentage of ESAVI notifications related to it (16.21%). The most common manifestations were a headache with an incidence rate (Ti) of 1.13 per 100,000 doses administered for the first dose, 1.54 per 100,000 doses for the second dose, and 21.34 per 100,000 doses for booster doses. Nausea had an incidence rate of 3.39 per 100,000 doses for the first dose and 3.05 per 100,000 doses for booster doses. Edema at the injection site had an incidence rate of 1.13 per 100,000 doses for the first dose and 6.10 per 100,000 doses for booster doses. Pain at the injection site had an incidence rate of 1.13 per 100,000 doses for the first dose and 6.10 per 100,000 doses for booster doses. None of these symptoms were reported during the second dose. Myalgia had an incidence rate only for booster doses, with 12.19 per 100,000 doses administered.

## 3. Discussion

According to the classification of ESAVI, it was found that in the population of Santarém, Non-Serious ESAVI (ESAVING) prevailed, with an incidence of 47.78 per 100,000 doses administered. Within the total of ESAVING cases, 3.06% were classified as immunization errors (Ti: 2.08). The remaining 14.0% were classified as serious ESAVI (ESAVIG) (Ti: 8.12 per 100,000 doses administered).

These findings corroborate with what is described in the Special Epidemiological Bulletin of the Ministry of Health of Brazil [14], which states that the majority of reported cases of ESAVI in the country were classified as ESAVING (national incidence of 53.8 per 100,000 doses administered). Outside Brazil, a study conducted by Castelo-Rivas et al. [15] with 600 residents in a locality in Ecuador—and Collado et al. [16] with healthcare workers in a hospital in Madrid, Spain—obtained similar results regarding the classification of ESAVI, with ESAVING being the most common in both studies [15,16].

Among the reported ESAVING cases in Santarém/PA, it is observed that the highest cumulative incidence is related to the Coronavac vaccine (Sinovac/Butantan), which differs from the national incidence, as in Brazil, according to the Ministry of Health, the highest cumulative incidence of ESAVING is associated with the Covishield vaccine (Astrazeneca/Fiocruz) with a Ti of 78.9 per 100,000 doses administered [14].

Regarding ESAVIG, the national incidence shows that the Coronavac vaccine (Sinovac/Butantan) is associated with the highest incidences of this event (6.6 per 100,000 doses administered); however, in Santarém/PA, these data are different, as ESAVIG cases are more related to the Covishield vaccine (Astrazeneca/Fiocruz) (18.11 per 100 doses administered). It is worth noting that among ESAVIG cases, Santarém had a higher Ti compared to the national incidence (Santarém Ti: 8.12 versus National Ti: 4.1); however, while the municipal incidence of death cases was 0, the national incidence was 1.3 per 100,000 doses administered [14].

The number of notifications classified as Immunization Errors (EI) was low, especially the number of EI cases with ESAVI, which had a Ti of 2.08 per 100,000 doses. According to the Technical Note No. 192/2022-CGPNI/DEIDT/SVS/MS, dated 22 July 2022, with the need for mass immunization to reverse the spread of the coronavirus, it is expected that Immunization Errors will occur, given that these campaigns are exhaustive and demanding for the professionals involved, factors that contribute to these events. The same document emphasizes that these cases can only be prevented through personnel training, adequate supply of equipment and supplies for vaccination, and proper human resource planning for mass vaccination, which allows the professionals involved to work with the best performance [17].

The data obtained regarding the time from vaccination to the onset of symptoms reveal that in Santarém/PA, the majority of cases had symptoms starting on the same day as immunization, which is consistent with the findings of Taborda et al. in a study conducted in Colombia describing that ESAVI for any COVID-19 vaccine has an onset within the first three days post-vaccination and is of short duration [18].

Regarding the sex variables of the most affected individuals, as indicated in the notifications, it was evident that the majority were females between 18 and 35 years old. This result is also found in the study by Francisco et al. [19] conducted in the municipality of Valença in the state of Rio de Janeiro, the study by Silva et al. [20] conducted throughout the state of Minas Gerais, and in the Epidemiological Bulletin of ESAVI produced by the State Center for Health Surveillance (CEVS) of Rio Grande do Sul [21] with data from the entire state.

This female predominance among cases is expected, considering the shortage of COVID-19 vaccines at the beginning of the vaccination campaign. The Ministry of Health defined priority groups, including healthcare professionals, who predominantly consist of female professionals in the field of nursing [22,23,24]. It should be noted that although these professionals were a priority for vaccination, they were also the ones who experienced the highest number of deaths in the workplace due to the COVID-19 pandemic, as they were in direct contact with patients and more vulnerable to the disease [25,26,27].

According to Gomes et al. [28], age is also a factor related to healthcare professionals, as many young individuals entered the healthcare workforce as recent graduates to meet the professional demands imposed by the pandemic. Additionally, as COVID-19 has a higher morbidity rate in older adults, the Federal Nursing Council (COFEN) recommended that older and/or comorbid healthcare professionals be removed from or transferred out of direct patient care activities to administrative tasks in order to protect these professionals [29,30].

Regarding the occurrence of ESAVI, it is possible to observe that there is a higher incidence with the first dose (Ti 84.84 per 100,000 first doses administered) compared to subsequent doses (Ti 0.90 per 100,000 doses administered). Studies on post-vaccination effects in the population report that post-vaccination events tend to be milder with the second dose of the vaccine, as the body has already generated an inflammatory response with the first dose, and the second dose acts as a booster, resulting in milder reactions [31,32,33]. Thus, as the municipality of Santarém/PA used the same vaccine for both the first and second doses and different vaccines for booster doses [34], the incidence of ESAVI associated with booster doses increased (Ti 1.17 per 100,000 booster doses administered).

In Santarém/PA, the majority of ESAVI notifications are related to the Covishield vaccine produced by Astrazeneca/Fiocruz. The most recurrent symptoms associated with this vaccine, as mentioned in the events, were fever, followed by headache and myalgia. The pharmaceutical package insert for Covishield classifies the side effects observed during clinical trials as “very common”, affecting more than 1 in 10 people, including sensitivity, pain, warmth, redness, itching, swelling, or bruising at the injection site, general feeling of being unwell, fatigue, chills or feverish sensation, headache, nausea, and joint or muscle pain. “common” side effects affecting up to 1 in 10 people include a lump at the injection site, fever, nausea, vomiting, and cold-like symptoms such as high fever, sore throat, runny nose, cough, and chills. “uncommon” side effects, which may affect up to 1 in 100 people, include dizziness, decreased appetite, abdominal pain, swollen lymph nodes, excessive sweating, itching, and skin rash [35].

Taking into account this classification, ESAVI events related to Covishield had 51.06% of manifestations classified as “very common”, 34.75% as “common”, and 3.54% as “uncommon”. It is worth noting that 10.63% of the manifestations found in the notifications were not described in the Covishield package insert, with diarrhea being a noteworthy symptom that is not mentioned in the vaccine’s label but is also found among the symptoms of ESAVI notifications in the state of Rio Grande do Sul, which corroborates the findings of this study [21].

The Coronavac vaccine produced by Sinovac/Butantan, which accounted for 29.72% of ESAVI notifications in the city of Santarém/PA, had the following most recurrent symptoms—headache, followed by myalgia, fever, vomiting, and diarrhea. The package insert for Coronavac also provides classifications of “very common”, “common”, and “uncommon”, differentiated by age groups and in which phase of the clinical trials the symptoms were observed (“Phase I/II adults 18–59 years and elderly over 60 years”, “Phase III adults 18–59 years”, “Phase III elderly over 60 years”) [36]. Based on ESAVI observed in one Phase III clinical trial in adults (18–59 years) and the elderly over 60 years in Santarém/PA, there were 40.32% “common” reactions, 33.87% “uncommon” reactions, and 24.19% “very common” reactions.

Hypotension (1.61%) was one of the symptoms not mentioned in the Coronavac package insert. Research on the Sinovac vaccine also highlighted other post-vaccination symptoms that were not reported in the vaccine’s label, such as the development of Evanescent Multiple White Dot Syndrome, reported in the study by Novais et al. [36], and the development of Pityriasis Lichenoides, evidenced in the study by Maestri et al. [37]. Both studies emphasize the importance of reporting ESAVI to ensure that healthcare professionals are prepared for any adverse events after vaccination and reiterate that the occurrence of these cases does not negate the benefit of using this vaccine in the population, as the benefits outweigh the risks [37,38].

Among the vaccines administered in the municipality, the Comirnaty vaccine produced by Pfizer/Wyeth had the lowest percentage of ESAVI reports (16.21%). The most common manifestations were headache, followed by myalgia and nausea, with higher incidences in booster doses. In a study conducted in Mexico [33], comparing the incidence of ESAVI between doses of Comirnaty, the majority of cases were associated with the first dose, which is consistent with the present research, as the booster dose was the first exposure of individuals to the Pfizer vaccine in the municipality, characterizing this finding [34].

Furthermore, unlike the package inserts for Covishield (Fiocruz/Astrazeneca) and Coronavac (Sinovac/Butantan), the Pfizer/Wyeth package insert includes two additional classifications besides “very common” (occurring in 10% of patients using this medication), “common” (occurring between 1% and 10% of patients using this medication), and “uncommon” (occurring between 0.1% and 1% of patients using this medication). These additional classifications are “rare” (occurring between 0.01% and 0.1% of patients), which includes acute facial paralysis as an event, and “unknown”—for which the laboratory could not estimate the occurrence rate among users of the vaccine—which includes severe allergic reactions (anaphylaxis) [39].

In Comirnaty, most of the events are classified as “very common” (33.33%), followed by “common” events (24.44%), “uncommon” events (6.66%), and atrophy at the injection site (2.22%), which was mentioned in the notifications but is not included in the Comirnaty package insert.

The data from this research demonstrate that the occurrence of ESAVI due to the COVID-19 vaccine in the municipality of Santarém/PA is not frequent compared to other studies that examine the number of ESAVI reports for COVID-19 vaccines. It is evident that the number of notifications (*n* = 74) in the municipality is lower than expected considering the number of doses administered and the diversity of vaccinated individuals in the study period [19,20,31]. In Brazil, as ESAVI reports are passively submitted by healthcare professionals when individuals who have received a vaccine return to the health facility with complaints of symptoms after vaccination, underreporting is common, resulting in a low number of notifications [40].

The effect of differences between vaccines on adverse events following immunization (AEFI) in Santarém/PA was significant. This study found that the majority of AEFI cases were classified as non-serious, affirming the safety of the vaccines used. The results revealed differences in the manifestations of AEFI among the different vaccines used, highlighting the importance of closely monitoring these reactions and providing updated information on vaccine-specific adverse events. For instance, the majority of ESAVI notifications in Santarém/PA were related to the Covishield vaccine produced by AstraZeneca/Fiocruz, with the most recurrent symptoms being fever, headache, and myalgia. In contrast, the Coronavac vaccine produced by Sinovac/Butantan accounted for a different percentage of ESAVI notifications, with headache, myalgia, fever, vomiting, and diarrhea being the most recurrent symptoms. The observed differences in AEFI manifestations across vaccines emphasize the need for continued vigilance and research to maintain public confidence in vaccination efforts and support mass vaccination strategies to combat the pandemic.

Additionally, when a case is reported, the challenges related to completing the notification forms in the e-SUS Notifica system must be considered, especially regarding the complexity of the requested information, which can influence the quality of the information and, consequently, the true situation of the case occurrence [41,42].

Thus, the limitation of this study lies in the use of secondary data, limited to specific information available in the notification form, which may contain inconsistencies beyond the researchers’ control as it was not the researcher who entered the notifications into the e-SUS Notifica system but rather various professionals from different locations.

This research, therefore, fosters the discussion about the importance of recording ESAVI resulting from COVID-19 vaccines, demonstrating their safety for the population and providing a basis for further studies with larger populations. Although the majority of events in this study were non-serious, there is a scarcity in the literature to contribute to the discussion. Thus, further studies in this area are crucial to increase the population’s confidence in vaccination.

## 4. Materials and Methods

This study was an epidemiological, descriptive study conducted using data from the e-SUS Notifica system in the city of Santarém, Pará, Brazil, from January 2021 to January 2022. All suspected cases of ESAVI related to COVID-19 vaccines administered in the city during the same period were analyzed, totaling 74 cases. Only individuals over 18 years of age were included.

Santarém is a city located in the interior of the state of Pará, comprising 46 neighborhoods and occupying a geographic area of 17,898 km^2^. It has an estimated population of 306,480 inhabitants, according to data from the Brazilian Institute of Geography and Statistics (IBGE) released in 2020 [43]. Santarém is the main city in the Baixo Amazonas health region and serves as a reference for healthcare services for adjacent municipalities. Due to this situation, the municipality must adopt dynamic mechanisms, especially in the healthcare sector, to serve the people residing in rural areas, forests, and rivers. These regions include a large concentration of traditional populations such as riverside dwellers, Indigenous people, and Quilombolas, who are more vulnerable to illness due to limited access to healthcare services, geographical distances, social vulnerabilities, inadequate transportation, and the need for the implementation of public health policies in the municipality [44].

In this research, suspected cases of ESAVI reported from January 2021 to January 2022 were analyzed using the e-SUS Notifica system, managed by the State Health Secretariat of Pará (SESPA). The variables analyzed for ESAVI cases included the administered immunobiologicals (Coronavac, Covishield, and Comirnaty), the type of event (non-serious, serious, immunization error, or immunization error with adverse event), case progression (full recovery without sequelae, full recovery with sequelae, under observation, not ESAVI, death, or other), and the time in days between immunobiological administration and symptom onset. ESAVI classification was based on the vaccine package inserts (very common reactions, common reactions, uncommon reactions, and rare reactions). The Incidence Rate (IR) per 100,000 doses administered was also calculated using the following formula:IR = ESAVI/(Number of doses administered) × 100,000 doses administered

The numerator considered the total number of COVID-19 vaccine ESAVI cases, and the denominator considered the number of doses of the same vaccine administered during the period (431,123). The number of doses administered was obtained from the National Immunization Program Information System (SI-PNI), which provides data on the National COVID-19 Vaccination Campaign for analysis [45].

Statistical Software for professionals (Stata), version 16.0, and Microsoft Office Excel 2016 were used for data analysis. The ESAVI estimates were presented as proportions (%) according to the administered immunobiological (Coronavac, Covishield, and Comirnaty), age group, and sex, as well as the type of event and case progression. The interval between vaccine administration and symptom onset was also analyzed.

This research was approved by the Ethics Committee of the State University of Pará, Campus XII, Santarém, under protocol CAAE 57418122.7.0000.5168.

## 5. Conclusions

In conclusion, this study analyzed the adverse events following COVID-19 vaccination in Santarém/PA and found that the majority of cases were classified as non-serious, reaffirming the safety of the vaccines used in the population. Although a higher incidence of AEFI was observed after the first dose, it is important to note that reactions tend to be milder with the second dose, highlighting the importance of completing the full vaccination schedule. Furthermore, the results revealed differences in the manifestations of AEFI among the different vaccines used, underscoring the need to closely monitor these reactions and provide updated information on vaccine-specific adverse events.

This research also emphasizes the ongoing need for adequate and comprehensive reporting of AEFI to improve epidemiological surveillance and ensure population safety. The scarcity in the literature of AEFI studies related to COVID-19 vaccines highlights the importance of further studies to provide a solid knowledge base and a better understanding of these adverse events. Based on the findings, it is evident that COVID-19 vaccines are safe, and the benefits of immunization far outweigh the risks associated with these adverse events. Continuing to monitor and report AEFI is crucial to maintain public confidence in the effectiveness and safety of vaccines and support mass vaccination strategies to combat the pandemic.

This study’s limitation lies in its reliance on secondary data from notification forms, which may contain inconsistencies not within the researchers’ control, as various professionals from different locations entered the data. This factor could influence the quality and accuracy of the information reported. Future research should aim to address these limitations by including larger populations and possibly integrating primary data collection to enhance the robustness of findings and further contribute to the discussion on vaccine safety and public health strategy effectiveness.

## Figures and Tables

**Table 1 pharmaceuticals-17-00304-t001:** Gender-based ESAVI notifications and doses administered for different vaccines in Santarém, Pará (January 2021–January 2022).

Gender	Vaccine	Notifications	Doses Administered	Incidence Rate (per 100,000 Doses)
**Female**	Coronavac	13	34,053	38.17
**Female**	Covishield	27	91,706	29.44
**Female**	Comirnaty	8	102,955	7.77
**Male**	Coronavac	9	29,660	30.34
**Male**	Covishield	13	84,960	15.30
**Male**	Comirnaty	4	87,789	4.55
**Overall**	Total	74	431,123	17.16

Source: e-SUS Notifica system, managed by the State Health Secretariat of Pará (SESPA).

**Table 2 pharmaceuticals-17-00304-t002:** Vaccine-specific ESAVI classifications and incidence rates in Santarém, Pará (January 2021–January 2022).

Vaccine	Doses Administered	ESAVI—Severe	Incidence Rate (ESAVI—Severe)	ESAVI—Non-Severe	Incidence Rate (ESAVI—Non-Severe)
**Coronavac**	63,713	0	0.00	63	98.88
**Covishield**	176,666	32	18.11	101	57.17
**Comirnaty**	190,744	3	1.57	42	22.02
**Total**	431,123	35	8.12	206	47.78

Source: e-SUS Notifica system, managed by the State Health Secretariat of Pará (SESPA).

## Data Availability

All data presented are available upon request from the corresponding author.

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
