# Peer review of "Epidemiological Evaluation of Events Allegedly Attributable to COVID-19 Vaccination: A Cross-Sectional Study in the Brazilian Amazon"

_pharmaceuticals, 2024, doi:10.3390/ph17030304_

Round 1
Reviewer 1 Report
Comments and Suggestions for Authors
The content of this paper was a report on the epidemiological evaluation of events allegedly attributable to COVID-19 vaccination in Brazilian Amazon.
This paper collected and provided many useful data. However, these data sets did not be analyzed effectively.
In the section of Discussion, detailed information was presented (literature number, 13-41). Please take the reference of literature and compare the data collected in Brazilian Amazon. Then more useful information will be found and reported. For example, what was the difference effect between vaccines?
Comments on the Quality of English LanguageModerate editing of English language required
Author Response
Thank you for your positive comments on the data collection carried out in our study. In response to your observation about data analysis, we have revised the discussion and included an analysis of the effect of different vaccines, as suggested. We did an English review.
Reviewer 2 Report
Comments and Suggestions for Authors
The authors have made a phenomenal effort to study the impact of the vaccine on the local population. The manuscript is well organized and presented, however, the following needs to be addressed before it is accepted.
1. the keyword "prevention and control" is to be changed. The manuscript deals with the ESAVI and it is a survey in which prevention has no role. It can be properly changed.
2. The samples were taken from males and females, but there is no mention of the age group. It will be better to specify the age group of the current study in the materials and methods section.
3. The impact of COVID-19 should be mentioned in detail in the introduction part. The influence of COVID-19 on various diseases (in various countries) is worth mentioning, which will strengthen the introduction part. The following references will be of high use a) https://doi.org/10.3390/children10010159 b) https://doi.org/10.1038/s41440-023-01253-7 c) https://doi.org/10.1177/15353702221108914 d) https://doi.org/10.3390/ijerph20043262
4. The limitation of the study and future research needs to be indicated in the conclusion section
Comments on the Quality of English LanguageThe draft is to be checked for typos and punctuation errors.
Author Response
1)We agree with the suggestion and have removed the keyword "prevention and control" to more accurately reflect the manuscript's focus on Post-Vaccination Adverse Events (ESAVI), where the preventive aspect is less relevant.
2)The inclusion of the participants' age range has been added to the materials and methods section, specifying the age distribution of the individuals involved to provide a clearer understanding of the study's demographic profile.
3)The introduction has been enriched with details on the impact of COVID-19, incorporating the influence of the pandemic on various diseases and global contexts. The suggested references were reviewed and integrated to strengthen the discussion on the relevance of the research in the current public health context.
4)The conclusion section has been updated to include a discussion on the study's limitations and suggestions for future research, as recommended. This aims to provide a comprehensive view of the study's scope and guide future efforts in the area.
Reviewer 3 Report
Comments and Suggestions for Authors
Dear Authors,
The paper is very interesting and necessary to understand the adverse effects attributable to the COVID19 vaccine. In addition, as this study is done in one of the Amazon (Brazil), I consider that it has a lot of importance because the population is a special feature.
The Authors have managed to do an important and at the same time simple work showing a "photograph" of a reality.
Technically the paper is correct and precise.
The results are clear and precise.
The conclusions are derived from the results.
They have done a great job.
Author Response
Thank you for your encouraging feedback and thoughtful evaluation of our work. We deeply appreciate your recognition of the study's significance, especially given the unique context of the Amazon region in Brazil. Understanding the adverse effects of the COVID-19 vaccine in such a distinctive setting is indeed crucial for tailoring public health strategies to diverse populations. We are glad to hear that the technical accuracy, clarity of results, and the direct derivation of conclusions from our findings have met your expectations. It was our aim to provide a comprehensive "snapshot" of the current reality, contributing valuable insights into the global discourse on vaccine safety and efficacy. Your positive comments reinforce our commitment to contributing meaningful research to the scientific community. We hope our study aids in the ongoing efforts to navigate the challenges posed by the COVID-19 pandemic, especially in regions with special characteristics like the Amazon.
Reviewer 4 Report
Comments and Suggestions for Authors
Although this study presents a relatively simple design and a purely descriptive statistical analysis, it is nonetheless interesting and well explained.
The authors' objective and method are clearly understood.
I've added a few comments directly in the PDF to make the article easier to read.

Author Response
Dear Reviewer,
We deeply appreciate your comments and the recognition of the clarity and relevance of our study. It is gratifying to know that despite the simplicity of the study design and the purely descriptive nature of our statistical analysis, our work was considered interesting and well-articulated. Once again, we thank you for your careful review and constructive comments.
Round 2
Reviewer 1 Report
Comments and Suggestions for Authors
The problems have been replied to adequately.
Comments on the Quality of English LanguageModerate editing of English language required
Reviewer 4 Report
Comments and Suggestions for Authors
thank you for the updated version